# Nuclear microenvironments modulate transcription from low-affinity enhancers

**Albert Tsai[1†]\*, Anand K Muthusamy[1‡], Mariana RP Alves[2], Luke D Lavis[1], Robert H Singer[1,3], David L Stern[1], Justin Crocker[1,2†]\***

[1]Janelia Research Campus, Howard Hughes Medical Institute, Ashburn, United States; [2]European Molecular Biology Laboratory, Heidelberg, Germany; [3]Department of Anatomy and Structural Biology, Albert Einstein College of Medicine, Bronx, United States

**Abstract** Transcription factors bind low-affinity DNA sequences for only short durations. It is not clear how brief, low-affinity interactions can drive efficient transcription. Here, we report that the transcription factor Ultrabithorax (Ubx) utilizes low-affinity binding sites in the *Drosophila melanogaster shavenbaby* (*svb*) locus and related enhancers in nuclear microenvironments of high Ubx concentrations. Related enhancers colocalize to the same microenvironments independently of their chromosomal location, suggesting that microenvironments are highly differentiated transcription domains. Manipulating the affinity of *svb* enhancers revealed an inverse relationship between enhancer affinity and Ubx concentration required for transcriptional activation. The Ubx cofactor, Homothorax (Hth), was co-enriched with Ubx near enhancers that require Hth, even though Ubx and Hth did not co-localize throughout the nucleus. Thus, microenvironments of high local transcription factor and cofactor concentrations could help low-affinity sites overcome their kinetic inefficiency. Mechanisms that generate these microenvironments could be a general feature of eukaryotic transcriptional regulation.
DOI: https://doi.org/10.7554/eLife.28975.001

**\*For correspondence:**
tsaia@janelia.hhmi.org (AT);
justin.crocker@embl.de (JC)

†These authors contributed equally to this work

**Present address:** ‡Division of Chemistry and Chemical Engineering, California Institute of Technology, Pasadena, United States

## Introduction

Genomic regions near coding genes, called enhancers, direct specific patterns of gene expression (*Spitz and Furlong, 2012*; *Reiter et al., 2017*; *Long et al., 2016*). Enhancers contain short DNA sequences that bind sequence-specific activating and repressive transcription factor proteins, and the integration of these positive and negative signals directs gene expression (*Crocker et al., 2016a*). Protein-DNA binding is often an ephemeral event; studies in mammalian cells demonstrate that transcription factors disassociate within seconds of binding to DNA (*Liu et al., 2014*; *Chen et al., 2014*; *Izeddin et al., 2014*; *Voss et al., 2011*; *Normanno et al., 2015*; *Morisaki et al., 2014*). Furthermore, recent studies in animals ranging from fruit flies to mammals have revealed that low-affinity DNA-binding sites are critical to confer specificity between related transcription factors having binding sites with similar DNA sequences (*Crocker et al., 2015*; *Farley et al., 2015*; *Farley et al., 2016*; *Lorberbaum et al., 2016*; *Antosova et al., 2016*; *Rister et al., 2015*; *Crocker et al., 2010*; *Crocker et al., 2016b*; *Tanay, 2006*; *Lebrecht et al., 2005*; *Rowan et al., 2010*; *Gaudet and Mango, 2002*; *Jiang and Levine, 1993*). Increasing the affinity of binding sites to more stably recruit transcription factors activates promiscuous gene expression (*Farley et al., 2015*; *Ramos and Barolo, 2013*), which leads to developmental defects. It is unclear how brief protein-DNA contacts can mediate efficient transcription from enhancers containing low-affinity binding sites.

One possible mechanism that could mitigate their kinetic inefficiency is to increase the local concentrations of transcription factors. At the scale of a single enhancer over a few hundred base pairs

long, multiple low-affinity binding sites for the same transcription factor in close proximity could increase the frequency of binding events by trapping the protein. Furthermore, interactions between transcription factors and cofactors with multiple binding sites within an enhancer could generate 'microenvironments' (*Reiter et al., 2017*) of high factor concentrations.

We have explored this problem using the *shavenbaby* (*svb*) locus, which contains multiple enhancers that drive specific patterns of *svb* gene expression in developing *Drosophila* embryos. Each of three characterized *svb* enhancers contains clusters of low-affinity binding sites for the Hox gene Ultrabithorax (Ubx). These enhancers also require a Ubx cofactor Homothorax (Hth) to function (*Crocker et al., 2015*). We have exploited robust transgenic tools in *Drosophila*, new fluorescent dyes, and new approaches to prepare embryos for microscopy to systematically perturb these *svb* enhancers and directly image the results at a sub-nuclear level. We find that microenvironments of high Ubx and Hth concentrations mediate transcription from low-affinity enhancers.

## Results

### Ubx is present in microenvironments of varying local concentrations

We first examined whether nuclei in *Drosophila melanogaster* embryos possess Ubx microenvironments by performing immunofluorescence (IF) staining in fixed embryos and high-resolution confocal imaging using Airyscan (Carl Zeiss Microscopy, Jena, Germany). We found that Ubx protein was not distributed uniformly, but rather exhibited regions of high and low Ubx intensities (*Figure 1A,B*). To observe Ubx distribution at higher resolution, we expanded the size of the embryos (*Tillberg et al., 2016*) by approximately four-fold in each dimension (*Figure 1C*). Nuclei of expanded embryos revealed distinct regions of high Ubx intensity separated by regions of low Ubx intensity. We observed, on average, $185 \pm 25$ (*n* = 12, three embryos) clusters per nucleus that were stronger than one-quarter of the maximum Ubx intensity within that nucleus (*Figure 1D,E*, and *Figure 1—figure supplement 1*).

One explanation for the observed distribution of Ubx is that transcription factors localize generally to accessible regions of the nucleus that have high levels of transcriptional activity. This mechanism, if shared by transcription factors in general, should yield Ubx distributions that mostly overlap with that of other transcription factors. Engrailed (En), a transcription factor unrelated to Ubx, displayed non-uniform sub-nuclear concentrations, but its distribution only partially overlapped with that of Ubx (*Figure 1—figure supplement 2A–C*, white regions indicate overlap). We similarly observed only partial overlap between Ubx and Even-skipped (Eve) (*Figure 1—figure supplement 2D–F*). Abdominal-A (AdbA), a paralog of Ubx that is expressed mainly in separate cells from Ubx and that has similar DNA-binding specificity as Ubx, was excluded from Ubx regions in the few nuclei where both were expressed (*Figure 1—figure supplement 2G–I*). These results indicate that the distributions of these transcription factors do not result from a shared mechanism that limits the distribution of all transcription factors to the same sub-nuclear regions.

We also examined whether Ubx simply occupies regions containing actively transcribed DNA. Both active RNA Polymerase II (Pol II, Ser5 phosphorylated CTD) and the methylated histone H3K4me3, which marks actively transcribed DNA, only partially overlapped with Ubx (*Figure 1—figure supplement 3A–F*). In contrast, the histone mark H3K27me3, which marks regions of repressed chromatin, displayed almost no overlap with the distribution of Ubx (*Figure 1—figure supplement 3G–I*). Thus, Ubx is not merely restricted to regions inside the nucleus that are available to transcription factors or to regions of high transcriptional activity.

### Ubx repeatedly binds to specific regions in nuclei of live embryos

To understand if the heterogeneous distribution of Ubx is dynamic or stable over the timescale of seconds to minutes, as well as to rule out the possibility that our observations of Ubx microenvironments are an artifact of the fixation protocol (*Teves et al., 2016*), we examined the spatiotemporal dynamics of single Ubx molecules in live *Drosophila* embryos. Single-molecule imaging has been mostly performed in cell lines previously because live-imaging studies of transcription factor dynamics in embryos requires overcoming several new challenges, including imaging at lower signal-to-noise ratios, compensating for rapid morphological changes during embryonic development, and determining how to deliver fluorescent dyes. We overcame these challenges by generating a

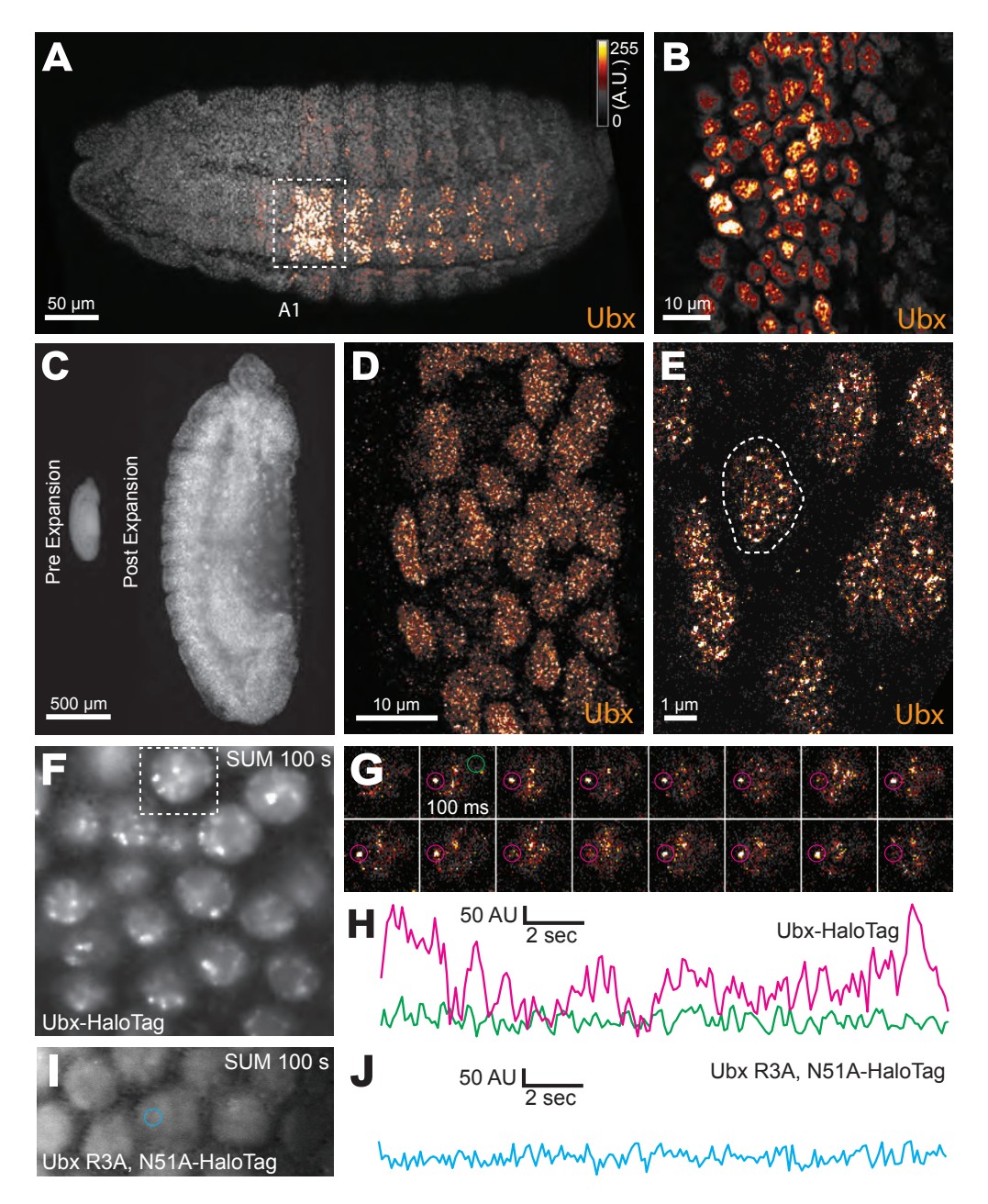

**Figure 1.** Ubx is present in microenvironments with varying local concentrations. (A) Stage 15 embryos stained for Ubx protein with a bounding box indicating a ventral region of abdominal segment one (A1). (B) Higher magnification, Airyscan image of the region indicated in panel (A). (C) Stage 15 embryo pre- and post-expansion. (D, E) Expanded stage 15 embryos stained for Ubx protein. The dashed line encircles a single nucleus in (E). (F, I) Projections of summed pixel intensity over 100 s from videos of nos::GAL4, UAS::HaloTag-Ubx for either a wild-type Ubx (F) or a binding deficient Ubx (I), imaged with $JF_{635}$ dye. (G) Sixteen individual, 100 millisecond video frames of the nucleus surrounded by a dashed box in panel (F). (H, J) Temporal traces of the signal intensity of the regions noted in panel (G) or (I). The color of each trace in (H) and (J) corresponds to the colors of the circles in panels (G) and (I), respectively. AU indicates Arbitrary Units of fluorescence intensity.

DOI: https://doi.org/10.7554/eLife.28975.002

The following figure supplements are available for figure 1:

**Figure supplement 1.** Quantification of Ubx microenvironments in single nuclei.

DOI: https://doi.org/10.7554/eLife.28975.003

**Figure supplement 2.** Ubx distribution compared to other transcription factors.

DOI: https://doi.org/10.7554/eLife.28975.004

*Figure 1 continued*

**Figure supplement 3.** Ubx distribution compared to general markers for transcriptional activity.
DOI: https://doi.org/10.7554/eLife.28975.005
**Figure supplement 4.** Control experiments for HaloTag-Ubx.
DOI: https://doi.org/10.7554/eLife.28975.006
**Figure supplement 5.** Additional nuclei from live imaging of Halo-Ubx.
DOI: https://doi.org/10.7554/eLife.28975.007
**Figure supplement 6.** Live imaging at higher concentrations of HaloTag-Ubx in late stage 6 embryos.
DOI: https://doi.org/10.7554/eLife.28975.008
**Figure supplement 7.** DNA-binding deficient Ubx is stable and localized in the nucleus.
DOI: https://doi.org/10.7554/eLife.28975.009

HaloTag-Ubx transgene that allowed precise control of fusion protein levels (*Figure 1—figure supplement 4A*) and coupling HaloTag-Ubx in vivo to new, strongly fluorescent dyes (*Grimm et al., 2017*). The transgene we built can be expressed either from a heat-shock promoter (*hsp70*) or from a *20x UAS* promoter by crossing with a GAL4 driver line.

Over-expression of the *HaloTag-Ubx* transgene by incubating the embryos at 30°C transformed anterior segments to the fate of more posterior segments, indicated by the presence of additional trichomes. This result indicates that the HaloTag-Ubx protein retains the expected Ubx behavior (*Figure 1—figure supplement 4D and E*). We then expressed HaloTag-Ubx from the *20x UAS* promoter with the *nos::GAL4* (*nanos* promoter driving *GAL4*) driver line, which drives HaloTag-Ubx expression in all cells at early developmental stages. We injected the HaloTag ligand of Janelia Fluor 635 (JF$_{635}$) (*Grimm et al., 2017*) into these live embryos. JF$_{635}$ is minimally fluorescent in solution but its fluorescence increases by over 100-fold when bound to a HaloTag protein, allowing the detection of labeled Ubx molecules against a background of dim freely diffusing dyes. The fluorescence intensity of labeled Ubx scaled with distance from the site of dye injection (*Figure 1—figure supplement 4B and C*), consistent with dye diffusion from the site of injection. To measure the time-averaged density of HaloTag-Ubx in specific locations of nuclei in live embryos in early stage 5, we calculated the summed intensity over 100 s (1000 frames at 100 ms per frame). We observed regions of Ubx signal (3-10x background) similar to the high-intensity clusters observed in fixed embryos (*Figure 1F*). We examined the dynamics of HaloTag-Ubx in nuclei by plotting fluorescence intensity over time (*Figure 1G and H* and *Figure 1—figure supplement 5*). We found that fluorescence signals over time changed in discrete up or down steps, indicating that individual HaloTag-Ubx molecules bind to specific nuclear domains with residence times on the order of a second before dissociation. Most unbound Ubx molecules move too quickly to be captured with the 100 ms exposure time; they move in and out of a diffraction-limited region in significantly less than 100 ms on average. These timescales are consistent with transcription factor-DNA binding dynamics measured in live-cell imaging experiments using mammalian cell lines (*Liu et al., 2014*; *Izeddin et al., 2014*; *Voss et al., 2011*; *Normanno et al., 2015*; *Morisaki et al., 2014*; *Gebhardt et al., 2013*). These repeated binding events produced the high intensities observed in the time-averaged projections and indicate that Ubx concentrates and remains within specific nuclear regions.

Observation of embryos at late stage 6 showed that total HaloTag-Ubx concentration continued to increase as the embryo ages (*Figure 1—figure supplement 6A and B*). Embryos at late stage 6 had nuclei containing high background concentrations of Ubx that masked single-molecule events, as well as displaying larger sites (>4 × 4 pixels) that constantly remained bright, possibly indicating the presence of multiple molecules or protein aggregation (*Figure 1—figure supplement 6C and D*). In contrast, the embryos observed during stage 5 did not contain areas that remained constantly bright, suggesting that we observed single molecule dynamics in stage 5 embryos.

To determine whether regions of high Ubx concentration depended on DNA binding, we performed the same experiments with a version of the *HaloTag-Ubx* transgene where Arg3 and Asn51 of the homeodomain were mutated to Ala (R3A and N51A), abrogating DNA binding (*Slattery et al., 2011b*). Both the wild-type and DNA-binding deficient Ubx were expressed and imported into the nucleus (*Figure 1—figure supplement 7A,B,D,E,G, and H*), suggesting that the protein is stable. In contrast, an unstable HaloTag-NLS construct (NLS from H2B) serving as a negative control, neither increased JF$_{635}$ fluorescence post injection nor became enriched into the

nucleus (*Figure 1—figure supplement 7C,F, and I*). The mutant HaloTag-Ubx (R3A N51A) did not display spatial heterogeneity and exhibited only extremely brief fluctuations in intensity inconsistent with transcription-factor DNA-binding events (*Figure 1I and J*). These results suggest that binding of Ubx to DNA is required to generate restricted nuclear distributions of Ubx.

## Transcriptionally active *svb* loci and enhancers correlate with regions of high Ubx concentration

The heterogeneous distributions of Ubx we observed are consistent with the hypothesis of nuclear 'microenvironments' (*Reiter et al., 2017*), whereby high local concentrations of transcription factors may drive transcription. Therefore, we examined whether these regions of high Ubx concentration co-localized with sites of active transcription. The *svb* gene is directly regulated by Ubx protein through binding of Ubx to low-affinity sites in multiple *svb* enhancers (*Crocker et al., 2015*). We marked sites of active *svb* transcription by fluorescence in situ hybridization (FISH) and compared the localization of actively transcribed *svb* loci to Ubx protein concentration (*Figure 2A and B*). We observed high local Ubx concentrations surrounding active *svb* transcription sites (*Figure 2C–F*). To quantify Ubx distributions around these sites, we calculated the radially averaged Ubx intensity as a function of distance $r$ from the point of maximum FISH intensity for each *svb* transcription site (*Figure 2G–I*). Ubx intensity was normalized to one at $r = 0$ (maximum FISH intensity) and averaged across all sites measured. To adjust for background fluorescence, we located the minimum intensity in the averaged Ubx distribution ($r = 2$–4 μm) and subtracted that value from the distribution. The first micrometer of the radially averaged 3D distribution is shown, with the shaded area representing the variance (*Figure 2J*). Within the first micrometer, *svb* transcription sites showed a relative enrichment of Ubx. Because these sites are on average within 200 nm of a local intensity maximum, Ubx intensity decreased monotonically away from the transcription sites, leading to a relatively constant variance after 200 nm. The normalized Ubx intensity after background subtraction at the site of *svb* transcription was $0.60 \pm 0.17$ ($n = 59$, four embryos, uncertainty is the variance of the background) and decreased approximately 250 nm away from the site. Thus, active *svb* transcription sites colocalized with areas of high Ubx concentration spanning approximately a few hundred nanometers.

If Ubx protein co-localizes with actively transcribed *svb* loci because Ubx drives *svb* expression, then we would expect that transcription at a locus not regulated by Ubx should not co-localize with high Ubx concentrations. Indeed, we observed that active transcription sites driven by a synthetic enhancer containing binding sites for a TALEA transcription factor (*Crocker et al., 2016a*; *Crocker and Stern, 2013*; *Crocker et al., 2017*) did not show Ubx enrichment on average despite wide fluctuations in Ubx levels, with a relative enrichment of Ubx at TALEA-driven enhancers of $0.02 \pm 0.63$ (*Figure 3A–C*, $n = 29$, three embryos). As these transcription sites are not close to maxima of Ubx intensity, the variance in these distributions incresed with distance from the site of transcription.

## Transcription sites of *svb* and *svb* enhancers co-localize

In numerous nuclei actively transcribing *svb* on the X chromosome, we observed what appeared to be two transcription sites within 200 nm of each other (*Figure 2—figure supplement 1A and B*). This indicates that the *svb* locus on homologous X chromosomes often co-localizes to the same Ubx microenvironment. There are several possible mechanisms that could explain this observation. We consider two broad classes of mechanism. First, a unique chromosomal signature specific to the region containing the *svb* locus could facilitate localization of homologous alleles to the same transcriptional microenvironments. Second, microenvironments contain distinct combinations of transcription factors and enhancers localize to the relevant microenvironments to enable transcription. To distinguish between these alternative hypotheses, we examined the spatial distribution of the native *svb* locus, located on the X chromosome, and a single *svb* enhancer driving *lacZ* expression which we placed on chromosome 3.

Double-FISH experiments revealed that the native *svb* locus and the ectopic *svb* enhancer co-localized often in nuclei in which both were transcribed (*Figure 2—figure supplement 1C and D*). In contrast, the transcription sites of *forkhead* (*fkh*, also on chromosome 3) did not colocalize with the *svb* locus (*Figure 2—figure supplement 1E*). The average distance between pairs of related transcription sites (*svb-svb, svb-7H, and svb-E3N*) within single nuclei is approximately 250 nm, near the

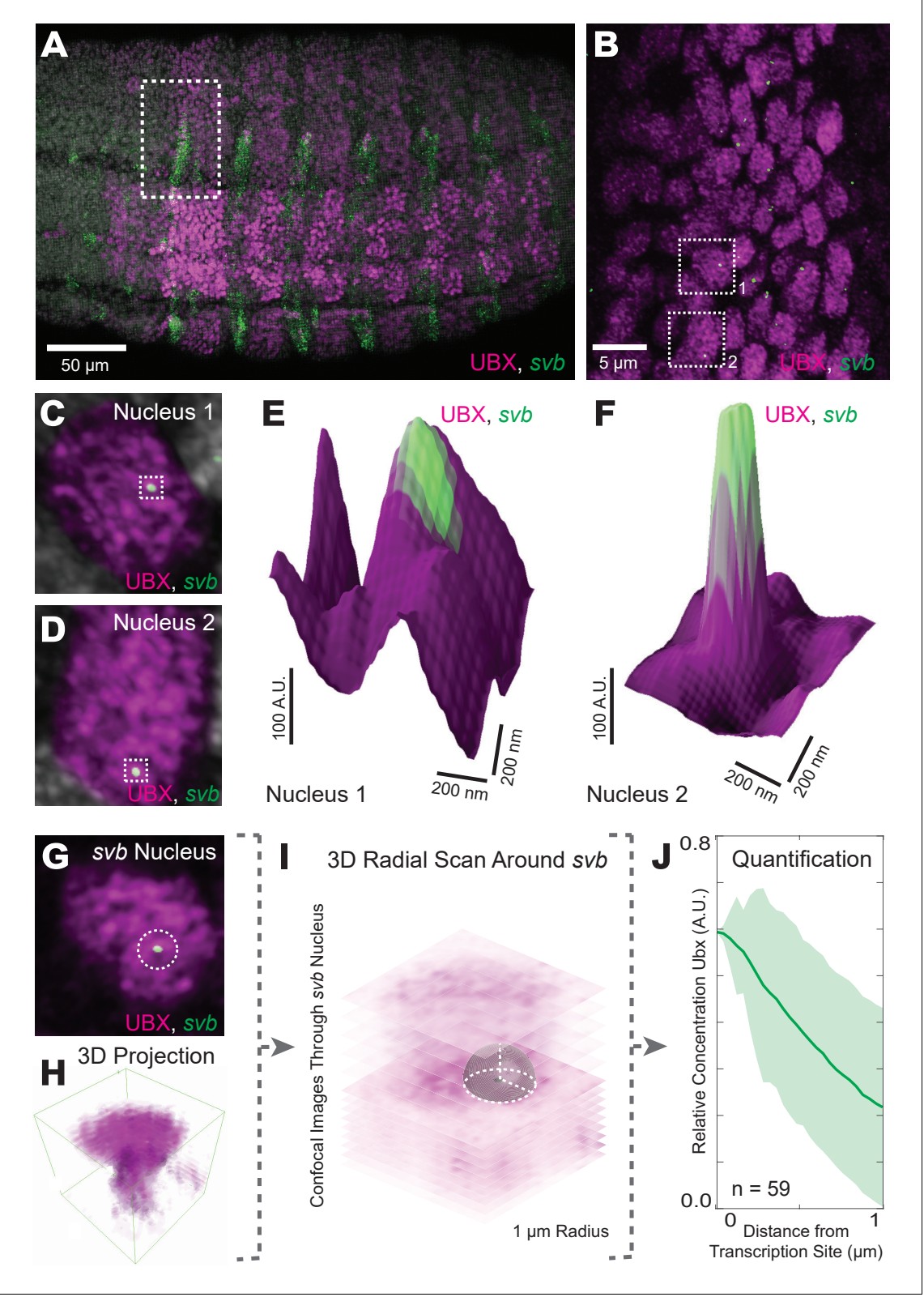

**Figure 2.** Transcriptionally active *svb* loci reside in regions of high Ubx concentration. (**A**) Embryos co-stained for both Ubx protein (magenta) and *shavenbaby* (*svb*) intronic mRNA (green). Bright spots of *svb* intronic nascent mRNA mark actively transcribed *svb* loci. Regions with high levels of both *svb* transcription and Ubx appear white (the sum of the two colors). (**B**) Higher magnification, Airyscan image of the region noted in panel (**A**), revealing sites of *svb* transcription (green). (**C, D**) Higher magnification, Airyscan images of the nuclei noted in panel (**B**). (**E, F**) 3D surface plots of the images in

*Figure 2 continued on next page*

*Figure 2 continued*

panels (C) and (D), centered on the sites of *svb* transcription (green), where height represents Ubx intensity. (G) A representative nucleus used for quantifying Ubx distribution around a *svb* transcription site. (H) 3D view of the confocal stack from the nuclei in panel (G). (I) Schematic outlining the method of Ubx quantification surrounding *svb* transcriptional sites. A 3D radial distribution of the average Ubx intensity on the surface of a sphere centered at the site of *svb* transcription was calculated. The gray sphere and white outlines is an example of the sphere with a radius *r* = 1 µm. (J) Quantification of the average relative concentration of Ubx and the distance from *svb* transcription sites (n = 59, see method supplements 'settings for extracting radially averaged distributions' for how relative concentration is computed). The shaded region indicates the variance. A.U. indicates Arbitrary Units of fluorescence intensity.

DOI: https://doi.org/10.7554/eLife.28975.010

The following figure supplement is available for figure 2:

**Figure supplement 1.** Transcription sites of minimal *svb* enhancers and the endogenous *svb* locus localize close to each other.

DOI: https://doi.org/10.7554/eLife.28975.011

resolution limit of AiryScan images (*Figure 2—figure supplement 1F*). On the other hand, *fkh* and *svb* transcription sites are on average 1 µm apart. These results indicate that related enhancers co-localize in transcriptional microenvironments independently of their chromosomal location. This suggests that transcription factor microenvironments are highly differentiated and that related enhancers often exploit the same transcriptional microenvironments.

## Manipulation of binding site number and affinity inversely changes the concentration of Ubx required to activate *svb* enhancers

The experiments described so far showed that the actively transcribed native *svb* locus co-localizes with local concentration maxima of Ubx in the nucleus. We wondered whether the position of actively transcribed enhancers within Ubx microenvironments depended on Ubx binding site affinity. To address this question, we examined transcription driven by the individual *svb* enhancers *DG3*, *E3N*, and *7* hr, each of which contains a cluster of low-affinity Ubx-binding sites and can independently drive transcription of a reporter gene when moved from their native location (*Crocker et al., 2015*). Transcription sites driven by these relocated enhancers also colocalized with regions of high Ubx concentration (*Figure 3D*). The relative Ubx enrichment for each of the three enhancers was 0.56 ± 0.16 for *DG3* (n = 61, three embryos), 0.51 ± 0.19 for *E3N* (n = 142, 11 embryos), and 0.68 ± 0.10 for *7* hr (n = 38, three embryos) (*Figure 3E–H,M,N*). These results indicate that low-affinity enhancers actively transcribed far from the native *svb* locus also co-localize with microenvironments of high Ubx concentrations.

Increasing the binding affinity of a site should increase its sensitivity to Ubx and allow transcriptional activation at lower Ubx concentrations. We found previously that replacing a single low-affinity Ubx site with one of a higher affinity led to higher levels of expression and sometimes drove promiscuous transcription (*Crocker et al., 2015*), suggesting that more stable Ubx-DNA interactions allowed higher transcriptional activation. Consistent with these previous results, we observed that increasing the affinity of a single low-affinity binding site in the *E3N* enhancer decreased Ubx enrichment near transcription sites to 0.44 ± 0.27 (*Figure 3I and J*, *E3N High Affinity*, n = 36, three embryos).

In contrast, we reported previously that deletion of low-affinity binding sites reduced transcription (*Crocker et al., 2015*). Removing some Ubx-binding sites should lower the effective affinity of the enhancer, and we hypothesized that this might result in transcription only when genes are localized to areas of higher Ubx concentrations. Consistent with this model, when we deleted two low-affinity sites in *E3N*, active transcription was observed in regions of increased Ubx enrichment (0.65 ± 0.18, *Figure 3K and L*, *E3N* Mut23, n = 62, five embryos). Deletion of two low-affinity Ubx sites from the *7* hr enhancer did not alter Ubx enrichment around transcription sites (0.63 ± 0.37, *Figure 3O and P*, 7H Mut23 n = 81, six embryos). But, deletion of three Ubx-binding sites in the *7H* enhancer increased relative Ubx enrichment, consistent with the pattern we observed for the *E3N* enhancer (0.91 ± 0.27, *Figure 3Q and R*, 7H Mut123, n = 52, eight embryos).

Across all manipulations, we observed an inverse correlation between binding site affinity and the distribution of Ubx intensities at transcription sites (*Figure 3—figure supplement 1*). Thus, the number of Ubx-binding sites and their affinities determine the response of *svb* enhancers to local Ubx

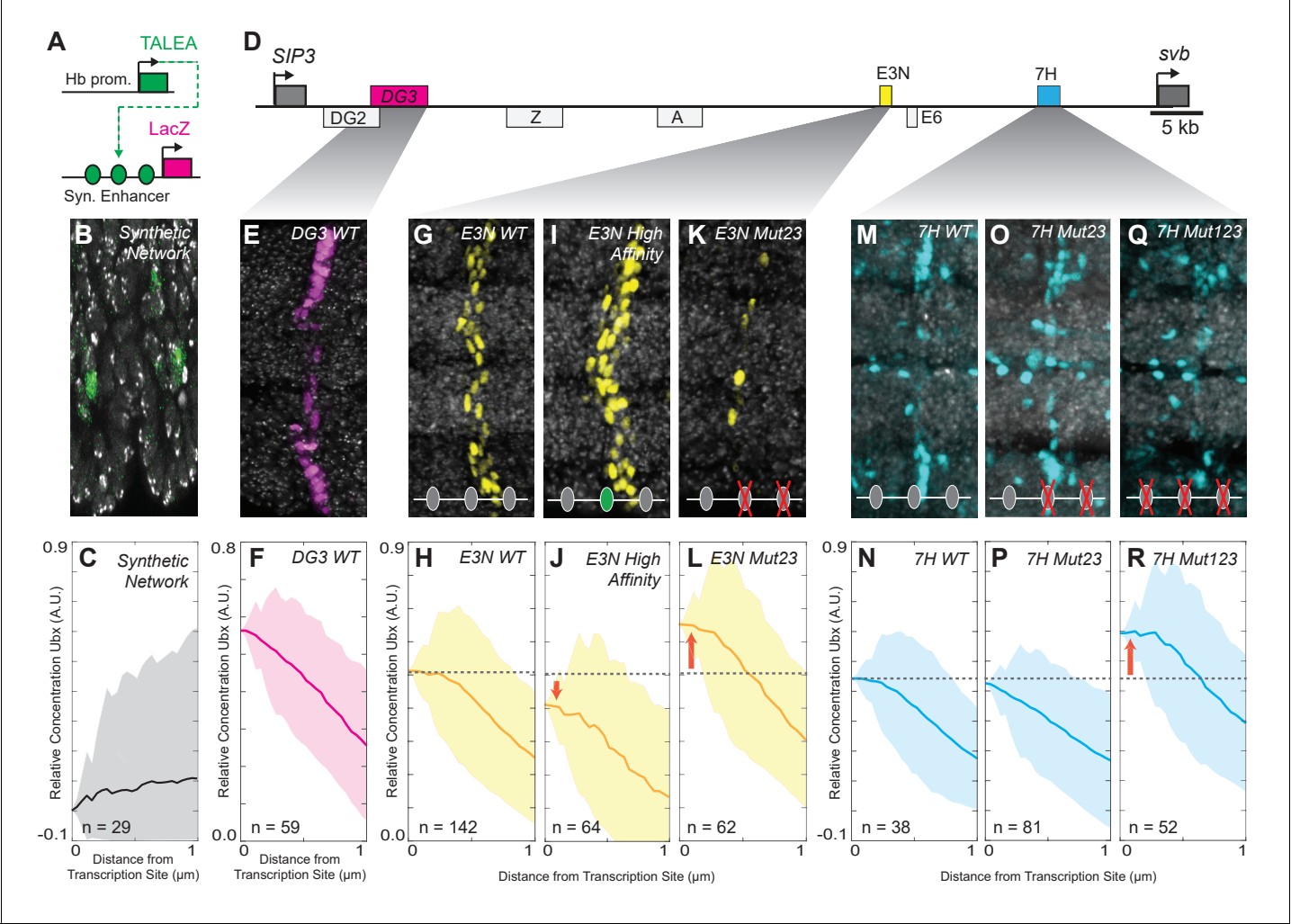

**Figure 3.** Manipulation of Ubx-binding site number and affinity alters the level of Ubx enrichment around *svb* enhancers. (**A**) Schematic of the synthetic TALEA transcription network driven by the Hunchback (Hb) promoter, indicating TALEA-binding sites with green circles. (**B**) Early stage 15 embryos carrying the TALEA synthetic network stained with an antibody against ß-Galactosidase. (**C**) Quantification of the relative concentration of Ubx based on the distance from synthetic network transcription sites. (**D**) Schematic of the *shavenbaby* locus, indicating embryonic *cis*-regulatory enhancers in boxes. The ventral embryonic enhancers *DG3*, *E3N* and *7H* are highlighted in magenta, yellow and blue boxes, respectively. (**E, G, I, K, M, O, Q**) Early stage 15 embryos carrying the reporter constructs *DG3-lacZ* (**E**), *E3N-lacZ* (**G, I, K**), or *7H-lacZ* (**M, O, Q**) stained with an antibody against ß-Galactosidase, with Ubx-Exd sites altered as indicated. (**F, H, J, L, N, P, R**) Quantification of the relative concentration of Ubx versus the distance from *svb* transcription sites. The shaded regions in panels (**C, F, H, J, L, N, P, R**) indicate the variance. A.U. indicates Arbitrary Units of fluorescence intensity.

DOI: https://doi.org/10.7554/eLife.28975.012

The following figure supplement is available for figure 3:

**Figure supplement 1.** Background-subtracted Ubx intensity distributions at the transcription sites for *E3N* and *7H* enhancers.

DOI: https://doi.org/10.7554/eLife.28975.013

concentration. Lower affinity enhancers require higher Ubx concentrations to drive transcription. Conversely, higher affinity enhancers can drive transcription at lower local Ubx concentrations.

Taken together, these data suggest that enhancers may be dynamically sampling local nuclear environments. A lower fraction of nuclei showing transcription from enhancers with binding site deletions (*Figure 3K,O,Q*) may occur because there are fewer areas of the nucleus in which peak Ubx levels are sufficient for weakened *svb* elements.

## The Ubx cofactor Homothorax (Hth) is co-enriched around transcription sites with Ubx

Co-factors can stabilize low-affinity binding interactions through cooperative and scaffolding interactions with transcription factors. A co-factor-dependent enhancer would require sufficient concentrations of both the factor and the co-factor to drive transcription. The homeodomain proteins Extradenticle (Exd)/Pbx and Homothorax (Hth)/MEIS (*Slattery et al., 2011b*; *Rieckhof et al., 1997*; *Ryoo and Mann, 1999*; *Lelli et al., 2011*) interact with Ubx during DNA binding, and Ubx and Hth regulate a partially overlapping set of genes (*Choo et al., 2011*; *Slattery et al., 2011a*). In vitro, Ubx requires Hth/Exd to bind to the low-affinity sites in *7H* and *E3N* (*Crocker et al., 2015*). In vivo, Hth deficiency led to the loss of expression for both *7H* and *E3N* (*Figure 4A–D*). Consistent with this requirement for both Ubx and Hth, Hth was co-enriched with Ubx around active transcription sites driven by *7H* or *E3N* (*Figure 4E–T*). The relative enrichment for Ubx and Hth, respectively, was $0.58 \pm 0.14$ and $0.41 \pm 0.16$ for *7H* ($n = 51$, seven embryos) and $0.66 \pm 0.13$ and $0.39 \pm 0.24$ for *E3N* ($n = 74$, five embryos). These results suggest that transcription from co-factor-dependent enhancers requires microenvironments that contain high concentrations of both transcription factors and their co-factors. This observation provides further support for the model that transcription factor microenvironments are present as multiple highly differentiated transcription domains containing unique combinations of transcription factors.

## Discussion

Biological systems often generate locally high concentrations of interacting molecules to increase the efficiency of biochemical reactions (*Dueber et al., 2009*; *Oehler and Müller-Hill, 2010*). This appears to be true also for transcription from low-affinity enhancers. Microenvironments (*Reiter et al., 2017*) of high local concentrations of transcription factors and their co-factors may circumvent the instability of low-affinity interactions by promoting more frequent DNA binding and cooperative interactions when enhancers are located within these domains (*Farley et al., 2016*) (*Figure 4U and V*). These microenvironments may be relatively stable domains generated by rapid dynamics of individual molecules. For example, we observed interactions between transcription factors and DNA on the timescale of seconds, with transcription factors continuously arriving to and departing from specific loci. From the perspective of gene expression, transcription likely occurs intermittently, switching on and off as the gene locus samples different nuclear regions. These rapid dynamics ensure that, once the gene locus moves outside of a microenvironment, or the conditions to form microenvironments are no longer satisfied, then the transcription factors needed to sustain expression quickly depart from low-affinity binding sites. In contrast, the fact that *svb* enhancers placed on the third chromosome often co-localized with the native *svb* locus on the X chromosome suggests that unique microenvironments may have relatively long half-lives. One challenge for the future is to determine how rapid dynamics of individual molecules generates apparently stable sub-nuclear domains.

Many mechanisms might work in concert to create these observed microenvironments. First, clustered binding sites for the same transcription factor (*Crocker et al., 2016b*) could lengthen the dwell time of proteins near enhancers and increase effective local protein concentrations (*Yao et al., 2006*; *Zhang et al., 2006*; *Elf et al., 2007*; *Kabata et al., 1993*; *Leith et al., 2012*; *Ruusala and Crothers, 1992*). Second, cooperative and scaffolding interactions between transcription factors and co-factors, each of which may bind independently to enhancers, can stabilize transcription factors at low-affinity sites (*Farley et al., 2016*; *Junion et al., 2012*). Finally, clustering of enhancers could trap transcription factors over longer length scales (*Noordermeer et al., 2014*; *de Laat and Duboule, 2013*; *Symmons et al., 2016*; *Williamson et al., 2016*; *Giorgetti et al., 2016*), perhaps generating the ~200 nm microenvironments that we observed. This last model is supported by recent findings that multiple promoters can share the same enhancer in a common local environment (*Fukaya et al., 2016*).

Transcription factor microenvironments may be a general feature of eukaryotic transcription, as supported by studies showing mouse and human cells exhibiting RNA polymerase II crowding (*Cisse et al., 2013*; *Cho et al., 2016*), transcription factors using local clustering to efficiently find their binding sites (*Liu et al., 2014*; *Izeddin et al., 2014*), and chromatin packaging in *Drosophila* cells generating distinct chromatin environments at the kilobase-to-megabase scale

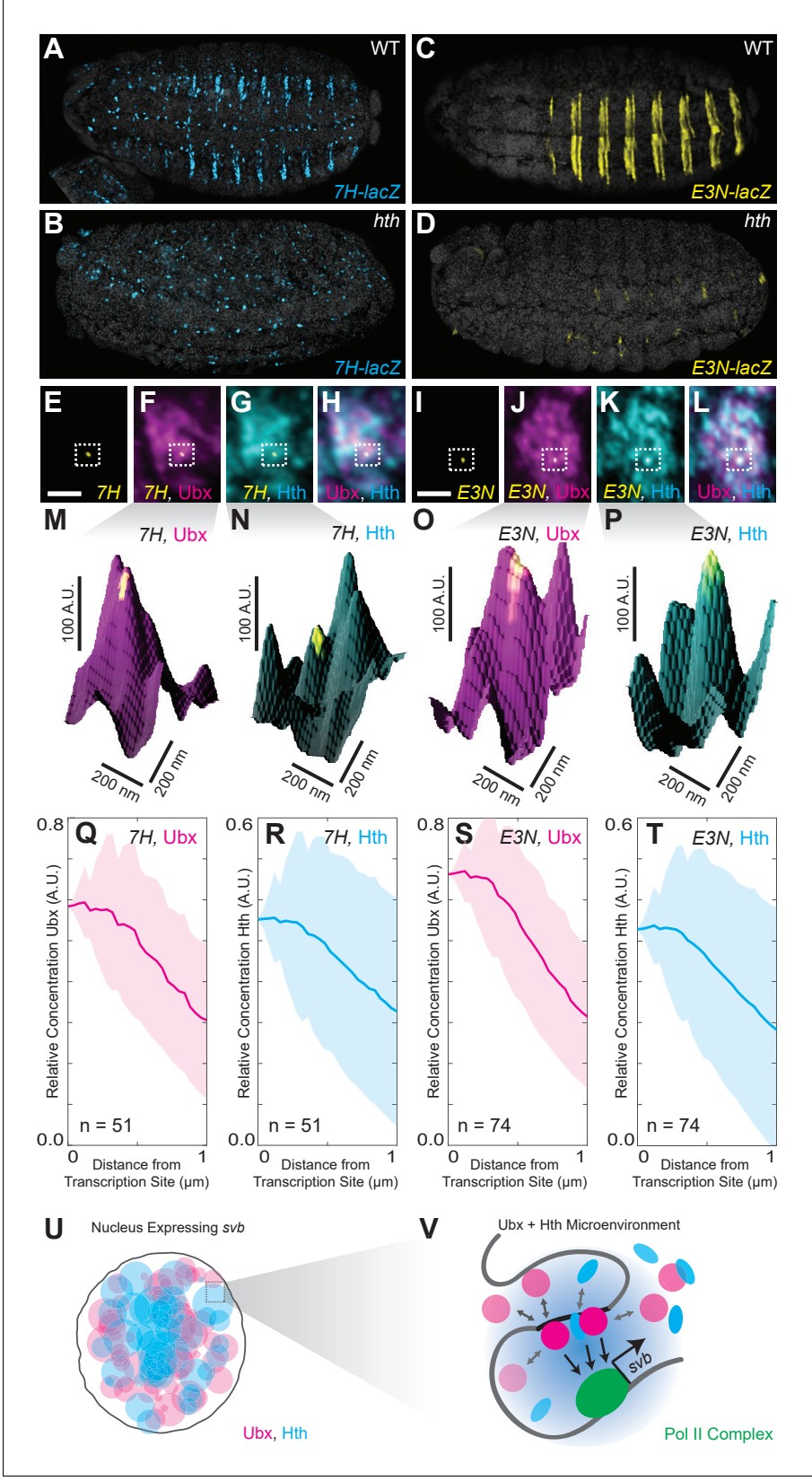

**Figure 4.** Ubx and its cofactor Hth are co-enriched around transcription sites. (**A–D**) Early stage 15 embryos with *7H-lacz* (**A–B**) or *E3N-lacZ* reporter constructs (**C–D**) stained with an antibody against ß-Galactosidase in either

*Figure 4 continued on next page*

*Figure 4 continued*

wild-type (WT) (**A and C**) or *hth*[P2] mutant embryos (**B and D**). (**E–H**) A nucleus displaying active transcription of the *7H-lacZ* reporter construct denoted by a bounding box (**E–H**) and co-stained for Ubx protein (**F**), Hth protein (**G**), or both Ubx and Hth proteins (**H**). (**I–L**) A nucleus displaying active transcription of the *E3N-lacZ* reporter construct denoted by a bounding box (**I–L**) and co-stained for Ubx protein (**J**), Hth protein (**K**), or both Ubx and Hth proteins (**L**). (**M–P**) 3D surface plots of the images in panels (**F, G, J, K**), centered on the sites of enhancer activity (yellow). The height of the plot is Ubx intensity in panels (**M**) and (**O**) and Hth intensity in panels (**N**) and (**P**). (**Q–T**) Quantification of the relative concentration of Ubx (**Q, S**) and Hth (**R, T**) versus distance from active enhancer sites. The shaded regions indicate the variance. A.U. indicates Arbitrary Units of fluorescence intensity. (**U, V**) A conceptual model showing nuclei with multiple regions of high local concentrations of Ubx or Hth (**U**) and high local concentrations of both Ubx and Hth that allow rapid ON rates (**V**, grey arrows) and collectively may recruit RNA pol II complexes.

DOI: https://doi.org/10.7554/eLife.28975.014

(*Boettiger et al., 2016*). Collectively, these findings are consistent with a phase-separated model of transcriptional regulation (*Hnisz et al., 2017*) whereby distinct microenvironments contain different combinations of proteins inside the nucleus. These localized regions impose a spatial constraint on the expression of genes, allowing transcriptional activation from enhancers only when they are physically in regions with the correct combinations of transcription factors and co-factors. Multiple enhancers acting as DNA scaffolds for protein binding could provide the anchoring interactions that form transcriptional microenvironments. These microenvironments would, in turn, provide a mechanism to allow both efficient and specific transcription from low-affinity enhancers.

## Materials and methods

### Preparing fixed *Drosophila* embryos

*D. melanogaster* strains were maintained under standard laboratory conditions. All enhancer constructs were cloned into the placZattB expression construct with a hsp70 promoter (*Crocker et al., 2015*). Transgenic fly lines were made by Rainbow Transgenic Flies Inc. *E3* and *7H* were integrated at the attP2 landing site. *DG3* was integrated at ZH-86Fb.

### Immuno-fluorescence staining of transcription factors and in situ hybridization to mRNA

Flies were reared at 25°C and embryos were fixed and stained according to standard protocols (*Crocker et al., 2015*). Primary antibodies were detected using secondary antibodies labeled with Alexa Fluor dyes (1:500, Invitrogen). In situ hybridizations were performed using DIG or biotin-labeled, antisense RNA-probes against a reporter construct RNA (*lacZ*) or the first intron of *svb* or *fkh*. DIG-labeled RNA products were detected with a DIG antibody: Invitrogen, 9H27L19 (1:200 dilution) and biotin-labeled RNA products are detected using a biotin antibody: Pierce, PA1-26792 (1:200).

The following primary antibodies for proteins were used at the indicated concentrations:
Ubx: Developmental Studies Hybridoma Bank, FP3.38-C (1:20)
Hth: Santa Cruz Biotechnology (dN-19), sc-26186 (1:50)
Eve: Developmental Studies Hybridoma Bank, 2B8-C (1:20)
AbdA: Santa Cruz Biotechnology (dN-17), sc-27063 (1:50)
En: Santa Cruz Biotechnology (d-300), sc-28640 (1:50)
RNA PolII RPB1 (Ser5 phosphorylated): BioLegend, (920304), (1:200)
Histone H3K27me3: Active Motif, 39157 (1:200)
Histone H3K4me3: Cell-signaling technology C42D8 (1:200)
LacZ: Promega anti-ß-Gal antibody (1:1000)

### Imaging fixed embryos with Airyscan

Fixed *Drosophila* embryos mounted in ProLong Gold mounting media (Molecular Probes, Eugene, OR) were imaged on a Zeiss LSM 880 confocal microscope with Airyscan (Carl Zeiss Microscopy,

Jena, Germany) using 3D Airyscan in SR mode to obtain images with 1.7-fold higher resolution compared to diffraction-limited confocal imaging (*Sheppard et al., 2013*) (method supplements: imaging setup for Airyscan). Images presented in the figures were processed with ImageJ (*Schindelin et al., 2015*).

## Expanding fixed embryos

To expand embryos, after fixation and staining, embryos were embedded into poly-acrylate gels and expended according to a previously published protocol (*Tillberg et al., 2016*) (method supplements: handling expansion gels).

## Imaging expanded embryos

Expanded gels containing embryos were imaged in 6-well glass bottom plates (Cellvis, Mountain View, CA) using a Zeiss LSM 800 confocal microscope (Carl Zeiss Microscopy, Jena, Germany) using standard settings (method supplements: imaging setup for expanded embryos).

## HaloTag-Ubx transgene construct for live imaging and overexpression assay

Transgenic fly lines containing HaloTag-Ubx under the control of both a *hsp70* and a *20x UAS* promoter was made by Rainbow Transgenic Flies Inc. The lines were made homozygous for the transgene.

## Preparing embryos for live imaging

Embryos resulting from crossing the homozygous line with the HaloTag-Ubx transgene with a *nos:: GAL4* driver line were injected following previously established protocols (*Rubin and Spradling, 1982*) with the HaloTag ligand of $JF_{635}$. Briefly, embryos were collected for 30 min at 25°C and placed in oxygen permeable Halocarbon 27 oil. The stock dye solution of 1 mM $JF_{635}$ with a HaloTag ligand in DMSO was diluted 1:100 into fly injection buffer and injected into the posterior end of the embryos. The embryos were then aged to stage 5 or late stage 6 and imaged in oxygen permeable Halocarbon 27 oil.

## Live imaging of *Drosophila* embryos

Injected embryos were imaged on a customized inverted Nikon Ti-Eclipse (Nikon Instruments, Tokyo, Japan) with the appropriate settings (method supplements: imaging setup for live embryos).

## Embryos for HaloTag-Ubx overexpression assay

Embryos from the homozygous HaloTag-Ubx transgene line were exposed to 30°C to induce the heat shock promoter and cuticle preps were prepared following previously established protocols (*Crocker et al., 2015*).

## Radially averaged distributions centered around transcription sites

To obtain the distributions of Ubx and Hth around a transcription site, the processed Airyscan stacks obtained from the Zeiss LSM 880 confocal microscope were analyzed in Fiji (*Schindelin et al., 2012*) using native functions and the 3D ImageJ Suite plugin (*Schmid et al., 2010*). Radially averaged distributions for individual transcription sites were computed using the 3D ImageJ Suite Plugin. Distributions for all sites were averaged and background offset in Matlab (MathWorks, Natick, MA) using a custom script (method supplements: settings for extracting radially averaged distributions).

## Method supplements

### Imaging setup for Airyscan

All Airyscan images were acquired using a Zeiss Plan-Apochromat 63x/1.4 Oil DIC M27 objective due to its well-characterized point spread function. First an embryo at the appropriate developmental stage (stage 15 for most embryos) and proper orientation was located. The band of mRNA expression in high Ubx regions of the first abdominal (A1) segment was then found. Within that band, areas containing transcription sites in nuclei of high Ubx expression were imaged. Images with both Ubx and Hth were acquired in the same manner by locating the proper area using the mRNA

and Ubx. When Ubx was imaged together with RNA polymerase II, a histone marker, or other transcription factors, Ubx expression levels were used to locate the region of interest.

The optimal setting suggested by Zeiss for the number of pixels in the x-y direction (40 nm per pixel) and displacement in the z-stack (190 nm) were used for all Airyscan images. The images from different fluorophores were acquired sequentially with the appropriate laser lines (405 nm, 488 nm, 561 nm, or 633 nm) and spectral filters. The laser power and gain were adjusted to maximize the signal to noise ratio within the dynamic range of the Airyscan detector. The acquired stacks were processed with Zen 2.3 SP1 (Carl Zeiss Microscopy GmbH, Jena, Germany) in 3D mode to obtain super-resolved images.

## Handling expansion gels

To allow easier handling of expanded gels, the gels containing embryos were cast into eight-well silicone isolators without adhesives (eight round chambers with a diameter of 9 mm and a thickness of 0.5 mm, Grace Bio-Labs (Bend, OR)) and allowed to polymerize. The gels were transferred into a six-well glass-bottom cell culture plate (Cellvis, Mountain View, CA) and expanded using ultrapure water containing 500 nM DAPI. Before imaging, the water was removed and the gel encased in 3% low melting temperature agarose (NuSieve GTG Agarose, Lonza Group Ltd, Basel, Switzerland), taking care not to allow the agarose to flow under the gel and float the gel away from the cover glass bottom. Water was then added back into the wells to prevent drying.

## Imaging setup for expanded embryos

A long working-distance water immersion objective, the Zeiss LCI Plan-Neofluar 25x/0.8 Imm Korr DIC M27, was selected for index-matching with the gel and its ability to image up to 400 μm above the surface of the coverslip. Stage 15 embryos in the correct orientation were located using the DAPI and Ubx staining. Regions of low to high Ubx expression were imaged sequentially using the appropriate laser lines (405 nm, 488 nm, or 561 nm) with the proper spectral filters. The laser power settings and the gain were selected to maximize signal to noise within the dynamic range of the detector. The full field of view of the microscope was imaged with 2048 × 2048 pixels and with a z-step of 1 μm. The final images presented were processed in ImageJ (*Schindelin et al., 2015*).

## Imaging setup for live embryos

All videos were collected under a Nikon CFI Plan Apo NCG 100X Oil NA 1.41 objective with an Andor iXon 897 EMCCD camera (Andor Technology Ltd., Belfast, UK). Embryos at stage 5 and late stage 6 in the correct orientation were found and imaged. We selected an area in the middle of the embryo with enough dye-labeled Ubx molecules to observe single molecules and we avoided regions close to the injection site to avoid oversaturating the camera (compare with Figure S5A-D where there are too many labeled Ubx). The samples were illuminated with a 633 nm laser to image the $JF_{635}$ tagged Halo-Ubx molecules with laser power and camera gain set to maximize signal from individual Ubx molecules without oversaturating the EMCCD detector. The 512 × 512 pixel videos were acquired at an exposure time of 100 ms per frame for up to 200 s. Images were processed using ImageJ to generate the time-averaged images and the intensity-over-time traces presented in the figures.

## Settings for extracting radially averaged distributions

To extract radially averaged protein distributions, we used Fiji to identify transcription sites inside nuclei by thresholding at a level that is roughly 50-fold above the background intensity. The center of a transcription site was defined as the pixel of maximum intensity in 3D in the mRNA channel inside a nucleus with high levels of Ubx expression. The radially averaged distribution out to a radius of 4 μm from transcription site for the transcription factor in 3D was computed using the 3D ImageJ Suite. The suite generates the distribution by computing the average intensity on the surface of a sphere with a radius $r$ from the center in three dimensions for all the values of $r$ ranging from zero to a desired outer limit (4 μm in this case).

The individual distributions from each transcription site were normalized to have the intensity at the center ($r = 0$) equal to 1. The distributions were averaged and background offset in Matlab. To adjust for background Ubx intensity outside of the nucleus, the entire averaged distribution was

offset by a constant value to bring the minimum intensity present in the distribution to zero to generate the distribution plots. The shaded area around the line represents the variance. The first µm of the distributions, where contributions from outside of the nucleus were minimal, are shown in the figures. The relative enrichment of Ubx or Hth for each enhancer variant is the intensity at $r = 0$ in the distribution and the cited uncertainty is the variance at the location of zero Ubx or Hth intensity (the site of minimum intensity before offsetting, between 2 and 4 µm from the transcription site).

The initial dataset for *7H* enhancers contained only a part of the deletion series. A subsequent dataset contained all the *7H* deletion mutants. The *7H* mutants present in both sets were compared and the distributions of Ubx intensity between the sets were found to differ by a multiplicative factor. When such factor was computed for each overlapping *7H* mutant present in both datasets, the results were similar, indicating that there was a systematic shift in background noise. This could have resulted from differences in embryo handling during fixation, antibody staining, and other steps in sample preparation. Other characteristics such as the functional form of the distributions between the two sets and the trends between *7H* mutants within each set remained unchanged after correcting for the difference in intensity. The wild-type *7H* data from the first set with a correction factor and the rest of the deletion series uncorrected from the second set were used to minimize the normalization employed.

## Acknowledgements

We thank Richard Mann, Timothée Lionnet, Paul Tillburg, and Brian English for advice and assistance on experimental design. We thank François Payre for advice on data presentation. We thank all members of the Stern and Singer labs for discussion. Albert Tsai is a Damon Runyon Fellow of the Damon Runyon Cancer Research Foundation (DRG 2220–15). Robert H Singer is supported by the 4D Nucleome Award U01-EB21236. Howard Hughes Medical Institute supported Albert Tsai, Anand K Muthusamy, Luke D Lavis, Robert H Singer, David L Stern, and Justin Crocker Mariana R P Alves and Justin Crocker are supported by the European Molecular Biological Laboratory (EMBL).

## Additional information

### Competing interests

Robert H Singer: Reviewing editor, *eLife*. The other authors declare that no competing interests exist.

### Funding

| Funder | Grant reference number | Author |
| --- | --- | --- |
| Damon Runyon Cancer Research Foundation | DRG 2220-15 | Albert Tsai |
| National Institutes of Health | U01-EB21236 | Robert H Singer |
| Howard Hughes Medical Institute | | Albert Tsai<br>Anand K Muthusamy<br>Luke D Lavis<br>Robert H Singer<br>David L Stern<br>Justin Crocker |
| European Molecular Biology Laboratory | | Mariana RP Alves<br>Justin Crocker |

The funders had no role in study design, data collection and interpretation, or the decision to submit the work for publication.

### Author contributions

Albert Tsai, Conceptualization, Data curation, Formal analysis, Validation, Investigation, Visualization, Methodology, Writing—original draft, Writing—review and editing, Conceived and designed the experiments, Executed the experiments, Analyzed the data; Anand K Muthusamy, Resources,

Provided reagents and design for live imaging experiments, Writing—review and editing; Mariana RP Alves, Investigation, Executed the experiments, Analyzed the data, Writing—review and editing; Luke D Lavis, Resources, Writing—review and editing, Provided reagents and design for live imaging experiments; Robert H Singer, Resources, Supervision, Funding acquisition, Writing—review and editing, Conceived of and designed the experiments; David L Stern, Conceptualization, Supervision, Funding acquisition, Investigation, Visualization, Methodology, Writing—original draft, Writing—review and editing, Conceived and Designed the experiments; Justin Crocker, Conceptualization, Supervision, Funding acquisition, Investigation, Visualization, Methodology, Writing—original draft, Writing—review and editing, Conceived and designed the experiments, Executed the experiments, Analyzed the data

### Author ORCIDs

Albert Tsai (iD) http://orcid.org/0000-0002-1643-0780
Mariana RP Alves (iD) https://orcid.org/0000-0002-0796-2101
Robert H Singer (iD) http://orcid.org/0000-0002-6725-0093
David L Stern (iD) https://orcid.org/0000-0002-1847-6483
Justin Crocker (iD) http://orcid.org/0000-0002-5113-0476

### Decision letter and Author response

Decision letter https://doi.org/10.7554/eLife.28975.019
Author response https://doi.org/10.7554/eLife.28975.020

## Additional files

### Supplementary files

• Source code 1. Matlab script to average radially averaged intensity distributions from individual transcription sites and offset background fluorescence.
DOI: https://doi.org/10.7554/eLife.28975.015

• Transparent reporting form
DOI: https://doi.org/10.7554/eLife.28975.016

### Major datasets

The following dataset was generated:

| Author(s) | Year | Dataset title | Dataset URL | Database, license, and accessibility information |
| --- | --- | --- | --- | --- |
| Tsai A, Muthusamy A, Alves M, Lavis L, Singer R, Stern D, Crocker J | 2017 | Data from: Nuclear microenvironments modulate transcription from low-affinity enhancers | http://dx.doi.org/10.5061/dryad.q96g6 | Available at Dryad Digital Repository under a CC0 Public Domain Dedication |

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
