## [Decision Letter]

Thank you for submitting your article "Nuclear microenvironments modulate transcription from low-affinity enhancers" for consideration by *eLife*. Your article has been favorably evaluated by Jessica Tyler (Senior Editor) and four reviewers, one of whom is a member of our Board of Reviewing Editors. The following individuals involved in review of your submission have agreed to reveal their identity: Robert P Zinzen (Reviewer #2); Stephen J. Small (Reviewer #3).

The reviewers have discussed the reviews with one another and the Reviewing Editor has drafted this decision to help you prepare a revised submission.

The reviewers of your manuscript found that the observations on the inhomogeneity of intranuclear Ubx transcription factor concentration and the activation of homologous enhancers with different Ubx affinities were highly significant and convincingly demonstrated. A strength of this study is that the authors use complementary techniques on fixed and living cells to observe the distribution of Ubx and other TF, and see similar trends. The use of *svb* regulatory elements with carefully "tuned" Ubx affinities and output is another powerful tool that reveals the importance of the different Ubx domains. Overall, the reviewers found that this study uses a set of advanced imaging methods to fundamentally reshape our view of the nuclear environment in the context of developmental gene regulation.

Essential points to be addressed:

1) There was some uncertainty whether the authors are proposing that the change in affinity of the Ubx motifs reshapes the intranuclear concentration gradients, or whether they are just responding differently to the existing inhomogeneities in the nucleus. The statement "Manipulation of binding site number and affinity changes the level of Ubx enrichment around *svb* enhancers" might be interpreted to mean that the selected enhancers are shaping the nuclear gradients. A clearer understanding of the authors' interpretation is necessary, and better statement of what they propose is establishing the different concentration of Ubx. Another related point: are the authors proposing that the enhancers get "stuck" in areas of appropriate concentration, or do they randomly sample the different intranuclear environments, activating when they are experiencing sufficient Ubx levels?

2) Pertinent to the above point, the in vivo halo-tagged imaging was found to be a very helpful complement to the analysis of fixed material, but the paper should do a better job of explicitly tying the results from the in vivo imaged spots to later work with the fixed, expanded embryos. The halo-tagged construct was not adequately described in the paper (only a brief outline in Figure 1—figure supplement 4). Finally, an important control for the in vivo expression was testing of a DNA-binding deficient form of Ubx, which did not show the inhomogeneous distributions, suggesting that DNA binding to something is essential for formation of the gradients. It was not clear that this negative control protein was expressed, however; a Western blot would demonstrate that the lack of signal is not due to trivial lack of stability.

3) An intriguing point of this paper is that as Ubx motifs are degraded, the active spots become localized only to regions of higher Ubx activity. In addition to the shown mean values that are higher for the low affinity enhancers, it is also clear from the embryo images that the weaker site enhancers are found in fewer nuclei altogether. Is this because the low affinity enhancer is able to respond to Ubx concentrations that are only found in a small percentage of nuclei? The authors are asked to show how the distribution of active enhancers in all nuclei changes as a function of Ubx motif affinity, as well as the mean values in nuclear regions where it is active. In addition, the mean levels of Ubx found in proximity to the active transcription loci have a very small variation at close range, and increase with increasing distance for a small range. Then the variation in concentration appears to be constant (and not continue to increase, as might be expected for searches of larger and larger spaces). The authors are asked to explain the shape of the Ubx distributions noted for the reporter constructs.

4) The reporter constructs with different Ubx affinities were derived from *svb* enhancer sequences. Do the reporters activate and colocalize with the endogenous *svb* locus in the fixed specimens? Double FISH with intronic probes should be able to discern this point.

---

## [Author Response]

Essential points to be addressed:1) There was some uncertainty whether the authors are proposing that the change in affinity of the Ubx motifs reshapes the intranuclear concentration gradients, or whether they are just responding differently to the existing inhomogeneities in the nucleus. The statement "Manipulation of binding site number and affinity changes the level of Ubx enrichment around svb enhancers" might be interpreted to mean that the selected enhancers are shaping the nuclear gradients. A clearer understanding of the authors' interpretation is necessary, and better statement of what they propose is establishing the different concentration of Ubx.

We agree that the original statement could imply that changing the enhancer architecture changes the distribution of Ubx in the nucleus. We have modified this statement to clarify that we do not expect this to happen. Our interpretation is that the mutated enhancers respond to existing Ubx concentration gradients within the nucleus. Even if the aggregated effects of transcription factor interacting with binding sites could shape their overall distribution, changing only one to three low affinity binding sites out of the thousands of possible Ubx binding sites in the genome of *D. melanogaster* is unlikely to change Ubx distribution in general.

Another related point: are the authors proposing that the enhancers get "stuck" in areas of appropriate concentration, or do they randomly sample the different intranuclear environments, activating when they are experiencing sufficient Ubx levels?

Our interpretation is that enhancers sample various areas of the nuclear environment and initiate transcription only when Ubx and cofactor concentrations are sufficient, but this is only our working model. In the future, we plan to characterize the dynamics of these interactions using additional live imaging strategies, but this technology is not yet fully operational.

2) Pertinent to the above point, the in vivo halo-tagged imaging was found to be a very helpful complement to the analysis of fixed material, but the paper should do a better job of explicitly tying the results from the in vivo imaged spots to later work with the fixed, expanded embryos.

It is not entirely clear what the reviewers are requesting here. We first observed microenvironments in fixed embryos and then expanded fixed embryos to examine the microenvironments at higher resolution. We then used live imaging to characterize the temporal dynamics of transcription factor binding and found that the dynamics were consistent with the observations of fixed specimens. We explicitly state that the live imaging was performed, in part, to test whether the observations of microenvironments in fixed specimens was an artifact of fixation. The consistency between the live-imaging and fixed specimens suggests that microenvironments are real. All following experiments were performed with fixed, unexpanded embryos.

The halo-tagged construct was not adequately described in the paper (only a brief outline in Figure 1—figure supplement 4).

We have added a description of the HaloTag-Ubx construct when we first introduce it in the Results, referencing the construct diagram in the figure. We also added a more complete description in the Materials and methods section. We also updated the diagram in Figure 1—figure supplement 4 to indicate the location of the actual HaloTag-Ubx sequence to further clarify its architecture. Additionally, we will deposit the construct to Addgene to enable use by the community.

Finally, an important control for the in vivo expression was testing of a DNA-binding deficient form of Ubx, which did not show the inhomogeneous distributions, suggesting that DNA binding to something is essential for formation of the gradients. It was not clear that this negative control protein was expressed, however; a Western blot would demonstrate that the lack of signal is not due to trivial lack of stability.

The DNA-binding deficient Ubx involves only two mutations to alanine in the DNA binding pocket and has previously been reported by Richard Mann and colleagues. With the binding deficient mutant, we still observed that the mutant Ubx is labeled using JF_635_ with a HaloTag ligand based on the bright fluorescent signals we observed post dye injection. Without a folded and functional HaloTag domain, the dye would have remained dark. The mutant Ubx is also selectively localized into the nucleus, indicating a functional NLS that has not been degraded. We still occasionally observed single-molecules of this Ubx inside the nucleus, but their apparent dwell-time is very short (less than 100 ms). These observations suggest that the mutant protein is expressed and stable so that it is not degraded immediately post translation and can be imported into the nucleus.

We also performed an experiment with just an H2B-derived NLS fused to the HaloTag, which is unstable, and we observed no fluorescence signal beyond background noise and auto-fluorescence. There was no enrichment in the nucleus. This control experiment demonstrates the effect of an unstable protein that is degraded immediately, in sharp contrast with the Ubx binding-deficient mutant. We have added Figure 1—figure supplement 7 and accompanying text in the Results section summarizing this point.

3) An intriguing point of this paper is that as Ubx motifs are degraded, the active spots become localized only to regions of higher Ubx activity. In addition to the shown mean values that are higher for the low affinity enhancers, it is also clear from the embryo images that the weaker site enhancers are found in fewer nuclei altogether. Is this because the low affinity enhancer is able to respond to Ubx concentrations that are only found in a small percentage of nuclei? The authors are asked to show how the distribution of active enhancers in all nuclei changes as a function of Ubx motif affinity, as well as the mean values in nuclear regions where it is active.

We agree with the interpretation that weaker enhancers require higher Ubx concentration to activate transcription. As a result, fewer microenvironments within a nucleus would contain sufficient concentrations of Ubx to activate transcription, lowering the probability that an enhancer exploring different areas within a nucleus would be transcriptionally active at any given moment. However, due to the heterogeneous distributions of Ubx in the nuclei of embryos at stage 15, the average concentration of Ubx over an entire nucleus is a poor representation of the specific conditions within microenvironments. While we observed that these weakened enhancers are active in microenvironments with higher Ubx concentrations, transcription sites are not preferentially located in nuclei of high average Ubx concentration. In fact, we did not observe transcription sites in most nuclei with some of the highest average levels of Ubx. It is known that other transcriptional inputs, both positive and negative, regulate *svb* transcription (Stern and Orgogozo, 2009). In this case, transcriptional repression may prevent *svb* expression at peak Ubx concentrations. It would be of interest to explore in greater detail the mechanisms of this activity as a part of our future work.

In addition, the mean levels of Ubx found in proximity to the active transcription loci have a very small variation at close range, and increase with increasing distance for a small range. Then the variation in concentration appears to be constant (and not continue to increase, as might be expected for searches of larger and larger spaces). The authors are asked to explain the shape of the Ubx distributions noted for the reporter constructs.

The convergence of the variance in Ubx intensity to zero at the transcription site is an effect of intensity normalization. The intensity of Ubx directly over the transcription site is normalized to one for each of the transcription sites. We average the distributions for all observed nuclei and then subtract the residual Ubx intensity 4 μm from the transcription site as the average background to generate the plots in Figure 2–Figure 4. Because the Ubx intensity at the transcription site is the point of normalization, its variance is zero.

The reviewers made an important observation concerning the relatively constant variance in the Ubx distributions after an initial increase. Because a transcription site may sit slightly off from the local maximum of Ubx concentration, the Ubx intensity distribution close to each site fluctuates within the first 200 nm (which is near the optical resolution of our images taken using AiryScan). This leads to an initial increase in variance as the distributions move away from the transcription sites. However, most transcriptions sites are close to a maximum of Ubx concentration, so Ubx intensity decreases monotonically after initial fluctuations. Because of the relatively uniform shapes of the individual distributions, the variance stops increasing after a short distance from the transcription site. We believe that there is no a priori reason that this must be the case. For example, the individual Ubx distributions around the transcription sites of the synthetic enhancer, which is not under the control of Ubx, fluctuate at random out to the full distance plotted in Figure 3 (out to 1 μm), leading to a variance that increases throughout the plot. We have added this observation into the Results.

4) The reporter constructs with different Ubx affinities were derived from svb enhancer sequences. Do the reporters activate and colocalize with the endogenous svb locus in the fixed specimens? Double FISH with intronic probes should be able to discern this point.

We conducted double FISH experiments and measured the distances of transcription sites between the endogenous *svb* locus (on the X chromosome) and other transcription sites (on chromosome 3). Enhancers related to *svb*, which are *7H* and *E3N*, preferentially colocalized close to the *svb* site in nuclei expressing both, despite being on different chromosomes. The average distance between transcription sites is about 200 nm. The transcription sites for *forkhead (fkh*), also on chromosome 3, did not preferentially colocalize with *svb*, with an average distance of 1 μm between transcription sites. In fact, nuclei showing two transcription sites from the *svb* locus (two bright spots next to each other) also have an average of 200 nm between those pairs of sites. We have added a new section in the Results and a new figure (Figure 2—figure supplement 1) describing this new result. We have also made additions in the Discussion highlighting this result. We would like to thank the reviewers for suggesting this experiment to improve our work.